# Academic global surgical competencies: A modified Delphi consensus study

Natalie Pawlak[1], Christine Dart[2], Hernan Sacoto Aguilar[3], Emmanuel Ameh[4], Abebe Bekele[5,6], Maria F. Jimenez[7], Kokila Lakhoo[8], Doruk Ozgediz[9], Nobhojit Roy[10], Girma Terfera[11], Adesoji O. Ademuyiwa[12], Barnabas Tobi Alayande[13], Nivaldo Alonso[14], Geoffrey A. Anderson[15], Stanley N. C. Anyanwu[16], Alazar Berhe Aregawi[17], Soham Bandyopadhyay[18,19], Tahmina Banu[20], Alemayehu Ginbo Bedada[21], Anteneh Gadisa Belachew[17], Fabio Botelho[22,23], Emmanuel Bua[24], Leticia Nunes Campos[25], Chris Dodgion[26], Michalina Drejza[27], Marcel E. Durieux[28], Rohini Dutta[29], Sarnai Erdene[30], Rodrigo Vaz Ferreira[31], Zipporah Gathuya[32], Dhruva Ghosh[33], Randeep Singh Jawa[34], Walter D. Johnson[35], Fauzia Anis Khan[36], Fanny Jamileth Navas Leon[37], Kristin L. Long[11], Jana B. A. Macleod[38,39], Anshul Mahajan[40], Rebecca G. Maine[41], Grace Zurielle C. Malolos[42], Craig D. McClain[43], Mary T. Nabukenya[44], Peter M. Nthumba[45,46], Benedict C. Nwomeh[47], Daniel Kinyuru Ojuka[48], Norgrove Penny[49], Martha A. Quiodettis[50], Jennifer Rickard[51], Lina Roa[52], Lucas Sousa Salgado[53], Lubna Samad[54], Justina Onyioza Seyi-Olajide[55], Martin Smith[56], Nichole Starr[9], Richard J. Stewart[57], John L. Tarpley[58,59], Julio L. Trostchansky[60], Ivan Trostchansky[61], Thomas G. Weiser[62], Adili Wobenjo[38], Elliot Wollner[63], Sudha Jayaraman[64]*

1 Tufts University, Boston, Massachusetts, United States of America, 2 Virginia Commonwealth University, Richmond, Virginia, United States of America, 3 Facultad de Medicina-Universidad el Azuay, Cuenca, Ecuador, 4 National Hospital Division of Paediatric Surgery, Abuja, Nigeria, 5 University of Global Health Equity, Butaro, Rwanda, 6 Addis Ababa University, Addis Ababa, Ethiopia, 7 Department of Surgery, Hospital Universitario Mayor Mederi, Universidad del Rosario, Bogota, Colombia, 8 University of Oxford, Oxford, United Kingdom, 9 Department of Surgery, University of California, San Francisco, San Francisco, California, United States of America, 10 The George Institute for Global Health, New Delhi, India, 11 Univ of Wisconsin, Madison, Wisconsin, United States of America, 12 Department of Surgery, Faculty of Clinical Sciences, College of Medicine, University of Lagos, Lagos, Nigeria, 13 Center for Equity in Global Surgery, University of Global Health Equity, Butaro, Rwanda, 14 University of São Paulo, São Paulo, Brazil, 15 Brigham and Women's Hospital, Boston, Massachusetts, United States of America, 16 Institute of Oncology, Nnamdi Azikiwe University Teaching Hospital, Nnewi, Nigeria, 17 Hawassa University, Hawassa, Ethiopia, 18 Nuffield Department of Surgical Sciences, Oxford University Global Surgery Group, University of Oxford, Oxford, United Kingdom, 19 Clinical Neurosciences, Clinical & Experimental Sciences, Faculty of Medicine, University of Southampton, Southampton, United Kingdom, 20 Chittagong Research Institute for Children Surgery, Chittagong, Bangladesh, 21 University of Botswana, Gaborone, Botswana, 22 Harvey E. Beardmore Division of Pediatric Surgery, Montreal Children's Hospital, Montreal, Canada, 23 Hospital das Clinicas da Universidade Federal de Minas Gerais, Belo Horizonte, Minas Gerais, Brasil, 24 Busitema University Mbale Hospital, Mbale, Uganda, 25 Faculty of Medical Sciences, Universidade de Pernambuco, Recife, Pernambuco, Brasil, 26 Medical College of Wisconsin, Milwaukee, Wisconsin, United States of America, 27 Specialty Trainee in Obstetrics and Gynaecology, Cambridge University Hospitals, Cambridge, United Kingdom, 28 University of Virginia, Charlottesville, Virginia, United States of America, 29 Program in Global Surgery and Social Change, Harvard Medical School, Boston, Massachusetts, United States of America, 30 Mongolian National University of Medical Sciences, Ulaanbaatar, Mongolia, 31 Universidade do Estado do Amazonas, Manaus, Brazil, 32 The Nairobi Hospital, Nairobi, Kenya, 33 NIHR Health Research Unit On Global Surgery, Christian Medical College, Ludhiana, India, 34 Stony Brook University, Stony Brook, New York, United States of America, 35 Loma Linda University, Loma Linda, California, United States of America, 36 Aga Khan University and Hospital, Karachi, Pakistan, 37 Hospital de Especialidades, Instituto Hondureño de Seguridad Social, San Pedro Sula, Honduras, 38 Kenyatta University, Nairobi, Kenya, 39 University of Pittsburgh School of Medicine, Pittsburgh, Pennsylvania, United States of America, 40 Global Surgery Fellow, WHO Collaboration Centre (WHOCC) for Research in Surgical Care Delivery in LMICs', Mumbai, India, 41 Department of Surgery, University of Washington, Seattle, Washington, United States of America, 42 College of Medicine, University of the Philippines Manila, Manila, Philippines, 43 Department of Anesthesiology, Critical Care and Pain Medicine, Program in Global Surgery, Harvard Medical School, Boston Children's Hospital, Boston, Massachusetts, United States of America, 44 Makerere

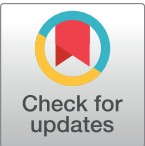

**Data Availability Statement:** All data is uploaded and available freely to the public: Jayaraman, Sudha. Academic Global Surgical Competencies: A Modified Delphi Consensus Study. Ann Arbor, MI:

Inter-university Consortium for Political and Social Research [distributor], 2022-12-21. https://doi.org/10.3886/E183623V1.

**Funding:** SJ has funding from the NIH Fogarty International Center (1R21TW011636-01A1). The funders had no role in study design, data collection and analysis, decision to publish, or preparation of the manuscript.

**Competing interests:** The authors have declared that no competing interests exist.

University College of Health Sciences, Kampala, Uganda, **45** Department of Surgery, AIC Kijabe Hospital, Kijabe, Kenya, **46** Department of Plastic Surgery, Vanderbilt University Medical Center, Nashville, Tennessee, United States of America, **47** Nationwide Children's Hospital, Columbus, Ohio, United States of America, **48** Department of Surgery, University of Nairobi, Nairobi, Kenya, **49** Branch for Global Surgical Care, University of British Columbia, Vancouver, Canada, **50** Hospital Santo Tomás, Santiago de Querétaro, Mexico, **51** University of Minnesota, Minneapolis, Minnesota, United States of America, **52** Department of Obstetrics & Gynecology, University of Alberta, Edmonton, Alberta, Canada, **53** União Educacional do Vale do Aço, Ipatinga, Minas Gerais, Brasil, **54** Interactive Research and Development (IRD) Global, Singapore, Singapore, **55** Lagos University Teaching Hospital, Idi-Araba, Lagos, Nigeria, **56** University of the Witwatersrand, Johannesburg, South Africa, **57** Global Initiative for Children's Surgery, Portland, Oregon, United States of America, **58** Department of Surgery, Faculty of Medicine, University of Botswana, Gaborone, Botswana, **59** Vanderbilt University, Nashville, Tennessee, United States of America, **60** Thoracic Surgery Department, Universidad de la República, Montevideo, Uruguay, **61** Hospital de Clínicas, Montevideo, Uruguay, **62** Department of Surgery, Stanford University, Palo Alto, California, United States of America, **63** Peter MacCallum Cancer Center and University of California, San Francisco, San Francisco, California, United States of America, **64** Department of Surgery, Center for Global Surgery, University of Utah Spencer Fox Eccles School of Medicine, Salt Lake City, Utah, United States of America

☯ These authors contributed equally to this work.
* Sudha.Jayaraman@hsc.utah.edu

## Abstract

Academic global surgery is a rapidly growing field that aims to improve access to safe surgical care worldwide. However, no universally accepted competencies exist to inform this developing field. A consensus-based approach, with input from a diverse group of experts, is needed to identify essential competencies that will lead to standardization in this field. A task force was set up using snowball sampling to recruit a broad group of content and context experts in global surgical and perioperative care. A draft set of competencies was revised through the modified Delphi process with two rounds of anonymous input. A threshold of 80% consensus was used to determine whether a competency or sub-competency learning objective was relevant to the skillset needed within academic global surgery and perioperative care. A diverse task force recruited experts from 22 countries to participate in both rounds of the Delphi process. Of the n = 59 respondents completing both rounds of iterative polling, 63% were from low- or middle-income countries. After two rounds of anonymous feedback, participants reached consensus on nine core competencies and 31 sub-competency objectives. The greatest consensus pertained to competency in ethics and professionalism in global surgery (100%) with emphasis on justice, equity, and decolonization across multiple competencies. This Delphi process, with input from experts worldwide, identified nine competencies which can be used to develop standardized academic global surgery and perioperative care curricula worldwide. Further work needs to be done to validate these competencies and establish assessments to ensure that they are taught effectively.

## Introduction

For years, surgeons, obstetricians/gynecologists, and anesthesiologists have been calling for equitable approaches to improving access to surgical care worldwide [1]. In 2015, the Lancet Commission on Global Surgery and the World Bank Disease Control Priorities Project, 3$^{rd}$ Edition, further highlighted the gross inequities in surgical services worldwide and the global burden of surgical disease [2, 3]. This propelled public health experts, who had initially

excluded surgery from global health discourse, to become major advocates for access to surgical and perioperative care in low- and middle-income countries (LMICs) [1, 2, 4]. The same year, the World Health Assembly, passed Resolution 68.15, which recognized surgical care as a key component of universal health coverage globally [5]. These events have motivated medical students, residents, and fellows from around the world in surgical specialties and in anesthesia to seek out formal educational experiences and scholarly pursuits that comprise the evolving field of academic global surgery.

However, despite this growing interest, the availability and quality of global surgical education programs have been limited and are almost entirely created by and for trainees from high-income countries (HICs) [6–10]. Indeed, there is no consensus on what global surgery is nor how it should be taught in local, regional, and global contexts. To date, no standardized or universal competencies exist in academic global surgery and perioperative care. In our recent systematic review, we found that, out of 119 publications on global surgery education or international surgery electives and curricula, only 18 (15%) mentioned any type of competency-based framework for trainees. All but one of the 18 publications were based in HICs and discussed programs set up for HIC trainees. Only four of the publications were open access. None explicitly cited "Health Equity and Social Justice" as a necessary competency, and few included "Social and Environmental Determinants of Health" which are clearly important competencies for equitable and just health care delivery. Additionally, there have been limited efforts to validate or gain consensus on global surgical curricula in collaboration with LMIC experts [10]. Thus, there is clear need to develop consensus around the fundamentals of academic global surgery to create curricula that can be accessed by the increasing numbers of trainees worldwide who are interested in this field. A consensus-based competency framework in global health education was published by Consortium of Universities for Global Health (CUGH) in 2015, but this did not include competencies related to surgical or perioperative disciplines. Therefore, this project aimed to address this gap in academic global surgical competencies by seeking to create a universal, consensus-based framework.

As a diverse and international group of surgeons involved in the American College of Surgeons (ACS) Operation Giving Back Education Sub-Committee, some of us recognized these challenges for the field of academic global surgery and formed an international team to develop the first geography-agnostic, consensus-based competency framework in academic global surgery and perioperative care [10]. We planned to do so through an iterative Delphi process, with an express emphasis on input from experts around the world. We now report on this process, which we hope will inform all foundational curricula in academic global surgery and perioperative care so that the next generation of surgeons and anesthesiologists understand the challenges, priorities, and values that should inform delivery of surgical services worldwide.

## Methods

### Ethics statement

This study was declared Exempt upon review by the University of Utah Institutional Review Board (IRB# 00135829). All participants in this study agreed to anonymous and voluntary participation in all rounds of the Delphi process through a formal written consent before the first-round survey.

### Participants

Following the systematic review conducted by many of the coauthors who are part of the ACS OGB Education Sub-Committee, a logistics team (SJ, NP, CD, DO, GT) was convened to take

on a global effort to define a framework of academic global surgical competencies for all learners interested in the academic components of global surgery, anesthesia, and perioperative care [10]. The logistics team mobilized an international task force (EA, AB, MFJ, KL, NR, HS), based on their interest in this topic, including involvement in the previous systematic review, expertise in this evolving field, and broad geographic leadership in academic global surgery and perioperative care [10]. The logistics team supported the conduct of the Delphi process including drafting the framework, creating the online survey, synthesizing responses and comments for the task force to review, drafting the final manuscript, and soliciting input from authors. This task force was responsible for creating a diverse list of experts to be invited to participate in this Delphi process, editing the first draft of the competency framework, evaluating the responses to each round, developing the final framework, and editing all drafts of the manuscript.

For this Delphi process, the logistics and task force teams defined an expert in academic global surgery and perioperative care as one or more of the following: 1) a committee member of an internationally-known professional surgical society, 2) a surgeon, anesthesiologist, or trainee who is a member of an internationally-known professional surgical or perioperative society, or 3) an individual nominated by a Delphi participant as someone who has significant and relevant experience in this topic area. Other criteria considered when selecting the expert panel included WHO region, country income level, surgical and perioperative specialty, and training level. Additionally, individuals needed "expert-level" experience in the field of academic global surgery and perioperative care, defined as a minimum of two years engaging in one (or more) aspects of the field (i.e., global surgery research, advocacy, surgical systems strengthening). Snowball sampling was used to expand the participants during the first round of the process. Medical students and residents with prior experience in academic global surgery and perioperative care who plan to continue their engagement in the field were also invited to participate to ensure that the voices of trainees were included in the competency development process.

## Process

We used a modified Delphi process, which is a systematic polling of the opinions of an expert panel knowledgeable on a given topic through iterative surveys to develop a final set of competencies (Fig 1) [11]. To start, an initial framework was developed by the logistics team based on existing competency-based curricula in global health and graduate surgical education and through a comprehensive literature review of competency-based global surgery curricula performed and published in a previous paper [10]. This was edited and reorganized by the task force and then distributed to the Delphi participants for further input through anonymous voting. Two rounds of responses were sought from the Delphi participants in accordance with consensus generating methodology previously used for global health education [12, 13]. A threshold of 80% consensus was established to retain any competency or objective, meaning that 80% of respondents voted "Agree" or "Strongly Agree" on a 5-point Likert scale for each competency and objective. An 80% consensus threshold was also used to determine whether to keep a competency or objective in the novice or advanced learning track, or both.

Each round included free text space for participants to provide feedback on any additions or changes to each competency or objective. They were also asked to assess if each competency was better suited for a novice or advanced track in academic global surgery and perioperative care. A "novice learner" was defined as one with no prior experience in global surgery and without previous exposure or access to educational resources on global surgery. An "advanced learner" was defined as someone who has had prior exposure to global surgery and plans to

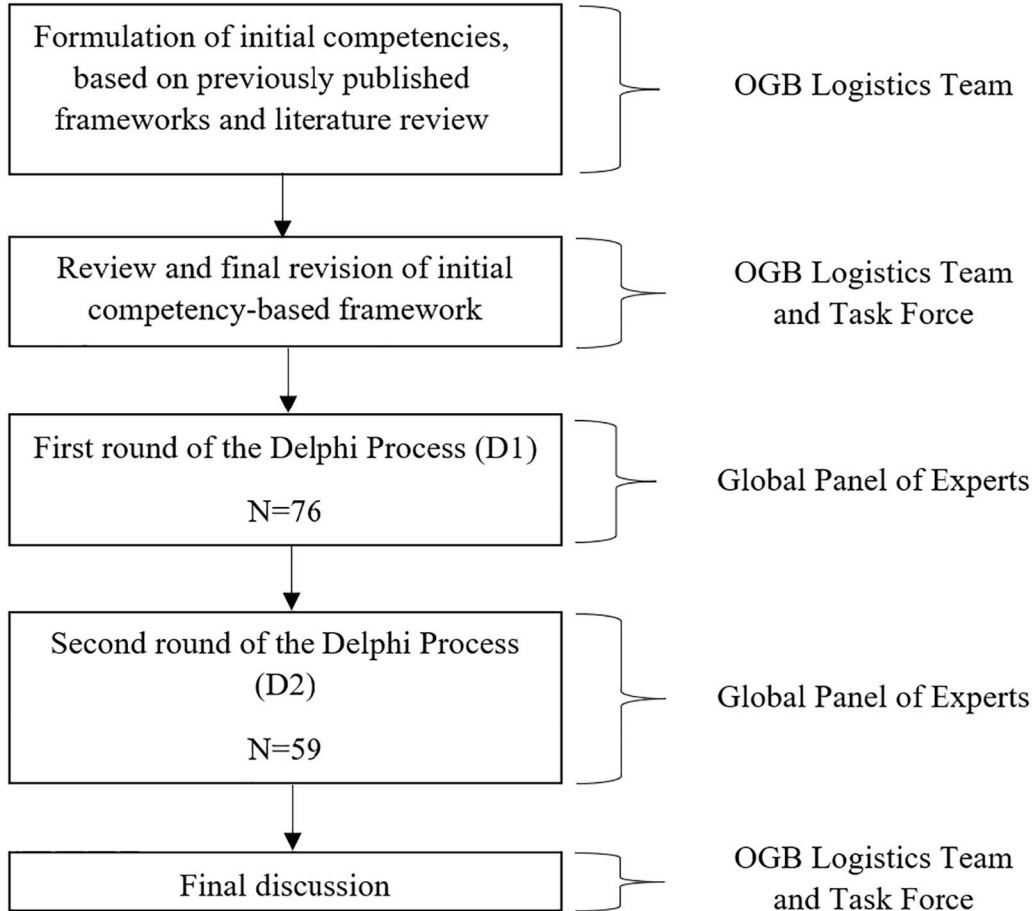

**Fig 1. Modified Delphi process for consensus driven development of global surgery competencies.** OGB: American College of Surgeons Operation Giving Back.

have continued involvement in global surgery. Participants were allowed to view comments by others from the previous round, but the responses were displayed anonymously to minimize bias created by undue influence among respondents. Competencies, and individual learning objectives comprising a competency, that did not meet the 80% threshold were removed from the framework. For competencies and objectives meeting the 80% threshold, the logistics team analyzed respondents' comments for major revision themes, consolidated the input, and circulated the updated competencies among the task force who voted to agree with or further modify the revisions prior to the second round. The modified competencies and objectives were then redistributed for the second round of voting and commenting, followed by synthesis and revisions to develop the final framework.

During each round, participants were sent one reminder email to complete the survey if they had not already done so. The logistics team and task force members were excluded from participating in the voting. A final discussion was held by email across the task force, in which members could bring attention to any issues or disagreement that they felt required further consideration before a final manuscript with the framework was drafted. All participants of both rounds of the Delphi process were invited to be co-authors on the manuscript along with the task force and logistics teams.

## Results

### Participants

The task force (EA, AB, MFJ, KL, NR, HS) provided initial feedback on a draft framework of 10 competencies and 34 sub-competency objectives which was then sent to participants in the Delphi Study. Countries represented by the task force included Nigeria, Ethiopia, United Kingdom, India, Colombia, and Ecuador. The overall Delphi process involved a total of 134 experts who were contacted, of which 76 participated in the first round (57% response rate), and of which 59 participated in the second and final round (78% response rate of first-round participants) for an overall response rate of 44%. Among the 59 individuals responding to both rounds, 37 (63%) were from LMICs. A total of 22 different countries were represented across five continents (Fig 2). A wide variety of surgical and perioperative specialties were represented by experts in the panel (Table 1). Medical students and residents with prior experience in academic global surgery and perioperative care consisted of a small portion of the respondents (9%, n = 5). Among the 54 experts who were post-graduate and senior faculty physicians, there was extensive experience in clinical practice.

### Consensus surrounding competencies in academic global surgery and perioperative care

The Delphi Process resulted in a final competency-based framework comprised of nine competencies and 31 sub-competency objectives. The competencies generating the greatest consensus after both rounds were Ethics and Professionalism in Global Surgery (100% consensus) and the Global Burden of Surgical and Perioperative Conditions and Injuries (98.3%

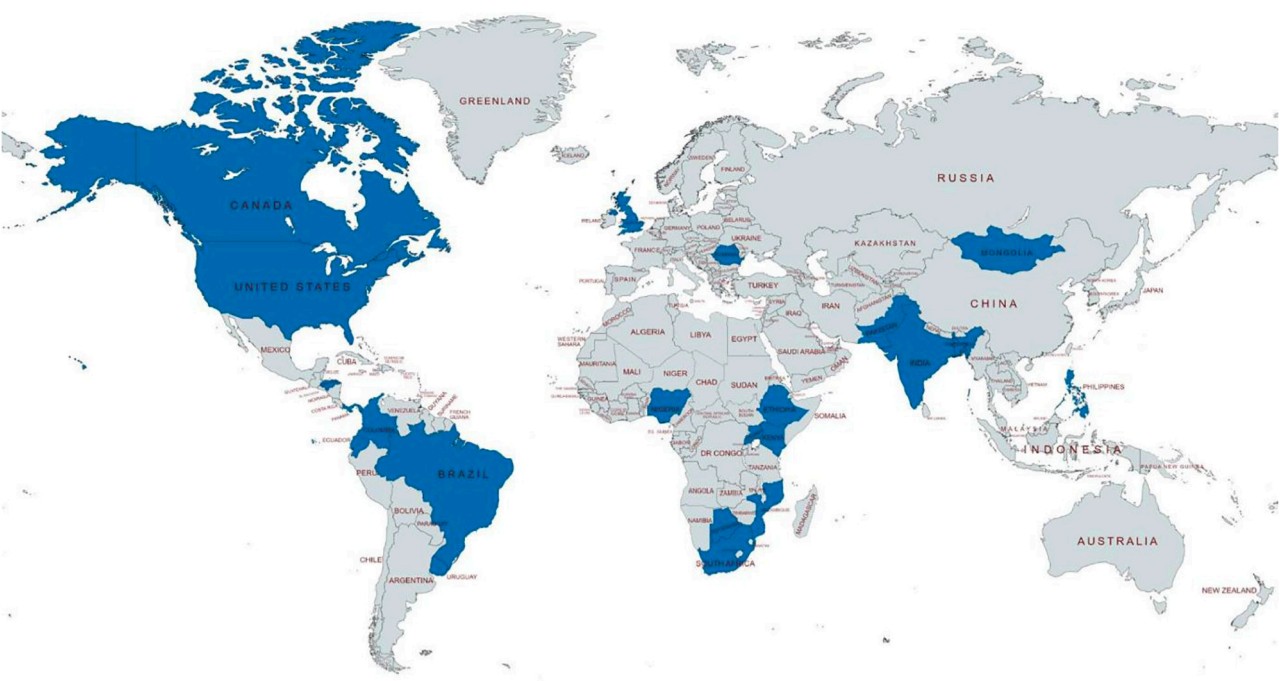

**Fig 2. World map depicting the location of Delphi respondents and task force members.** Experts from countries highlighted in blue participated in the modified Delphi Process. This work is licensed under a Creative Commons Attribution-ShareAlike 4.0 International License from Mapchart. https://www.mapchart.net/world.html.

**Table 1. Demographic and clinical training characteristics of Delphi participants.**

| Characteristic | Percentage of Respondents (Total n = 59) |
|---|---|
| *Demographics* | |
| LMIC | 63% (37/59) |
| HIC | 37% (22/59) |
| *Training Level and Experience in Clinical Practice* | |
| Medical Student or Junior Trainee | 9% (5/59) |
| 0–10 years | 35% (21/59) |
| 11–20 years | 22% (13/59) |
| 21–30 years | 17% (10/59) |
| >30 years | 17% (10/59) |
| *Clinical Specialty:* | |
| General Surgery | 32% (19/59) |
| Pediatric Surgery | 17% (10/59) |
| Trauma and Surgical Critical Care | 17% (10/59) |
| Anesthesiology | 13% (8/59) |
| Plastic Surgery | 5% (3/59) |
| Neurosurgery | 3% (2/59) |
| Orthopedic Surgery | 2% (1/59) |
| Obstetrics-Gynecology | 2% (1/59) |

consensus). (Table 2). The competency on ethics and professionalism also contained one of the five highest ranking sub-competency objectives in terms of agreement (98.3% consensus), which focused on understanding basic ethical principles and defining the common ethical challenges to delivering surgical and perioperative care in diverse cultural/political/economic settings, with a focus on practicing within one's experience level.

During voting rounds and revisions, two competencies had the least consensus and did not reach the quantitative 80% consensus for inclusion—Patient Safety and Quality Improvement, and Professional Practice—which were focused on clinical skills and practices. Respondents reported that these topics did not need separate competencies because: 1) they were included in other sub-competency objectives, 2) these competencies could not be universal and geography-agnostic because they would have to, by definition, be context specific, and 3) the competency framework focused on academic competencies which are distinct from clinical competencies. For these reasons, these two competencies were not included in the final version of the framework.

## Common qualitative themes emerging through the Delphi process

Qualitative feedback was received through the Delphi process as 186 unique comments in round 1, 88 in round 2, and 56 in the final manuscript review. Many of the responses contained detailed sub-components to review. There was additional discussion by email among the task force and logistics teams on the initial draft of the competencies, after each round and after the final manuscript review. Throughout the consensus process, several recurrent themes emerged which are illustrated below through quotes from respondents describing their perspectives.

Respondents noted the benefit of a structured curriculum to both trainees and institutions and especially emphasized the expertise and leadership of LMIC professionals and peer mentorship, specifically for Competency 8: Leadership and Competency 9: Research Equity and Publication.

**Table 2. Consensus-based core competencies and objectives in academic global surgery and perioperative care developed through a Delphi process (no specific order).**

| | Competency | | Objective | Consensus Threshold ≥80% | Novice Track | Advanced Track |
|---|---|---|---|---|---|---|
| 1 | **Global Burden of Surgical and Perioperative Conditions and Injuries** <br><br> **Consensus to keep competency: 98.3%** | a | To understand general public health concepts including epidemiology, measures of morbidity and mortality, and multiple determinants of health in relation to the global burden of surgical disease (GBSD) within and across various geographical areas. | 96.6% | X | X |
| | | b | To gain familiarity with major public health efforts to reduce the GBSD (e.g., Global Surgery Indicators, Sustainable Development Goals in relation to surgery) on a macro- and regional/ ethnic-specific level and the challenges these initiatives face. | 94.9% | X | X |
| | | c | To develop a fundamental understanding of analyzing, interpreting, and auditing public health data on surgical and perioperative morbidity and mortality across different settings (e.g. WHO IRTEC, Globocan, local surveillance data). | 89.8% | | X |
| 2 | **Globalization of Health, Health Systems, and National Surgery Plans** <br><br> **Consensus to keep competency: 94.9%** | a | To identify how global trends in health care practice, commerce and culture, multinational agreements, and multinational organizations contribute to the quality and availability of surgical care. | 91.5% | X | X |
| | | b | To explain how travel, trade, natural disasters, and armed conflict contribute to surgical and perioperative problems (e.g., Tobacco products and cancer epidemiology, Trauma patterns and refugee status secondary to war). | 93.2% | | X |
| | | c | To define core principles of healthcare economics in surgical services delivery and analyze various models of financing healthcare and national health plans. | 86.4% | X | X |
| | | d | To define and introduce the concept of National Surgical, Obstetric and Anesthesia Plans (NSOAPs). | 88.1% | X | X |
| | | e | To understand surgical systems strengthening as an indicator of preparedness for pandemics and natural disasters. | 86.4% | | X |
| | | f | To identify the roles of major social, civic, and political entities (i.e., district-level health systems, national health systems, Ministries of Health—as well as sociopolitical influences such as capitalism) impacting the development of local and regional surgical practices and inequities. | 88.1% | | X |
| 3 | **The Impact of Social and Environmental Determinants of Health and Surgical Care** <br><br> **Consensus to keep competency: 96.7%** | a | To define cultural influences and historical contexts and to understand how these factors impact perceptions, stigma, and belief systems surrounding surgical care. | 91.5% | X | X |
| | | b | To list the major social, political economic, and environmental determinants of health leading to regional (local and macro-level) disparities in surgical access and outcomes. | 94.9% | X | X |
| | | c | To understand how resource limitations (including limitations in workforce, infrastructure, medical equipment, etc.) impact practice patterns in various settings | 98.3% | X | X |
| 4 | **Strengthening Surgical Systems Capacity** <br><br> **Consensus to keep competency: 96.6%** | a | To explore and understand the surgical ecosystem inclusive of anesthesia, nursing, allied health, and all hospital systems related to procurement, supply chain, etc. and to be able to identify the cause of major barriers to creating or sustaining an efficient and effective system. | 98.3% | X | X |
| | | b | To demonstrate the impact of a strong surgical system on the entire healthcare system of a community and to identify the successful components of creating and maintaining the surgical ecosystem (including all care providers, referral systems, and decision-making parties involved in surgical care access locally). | 94.9% | X | X |
| | | c | To establish the importance of monitoring and evaluation (M&E) as a key component of surgical services and to understand the components of implementing M&E, which should be defined and led by local, national, or academic institution leadership. | 91.5% | X | X |
| | | d | To demonstrate the planning, implementing, and evaluating of evidence-based quality improvement programs, informed and led by local clinical QI teams, that deliver improved and sustainable surgical outcomes. | 91.5% | | X |

*(Continued)*

**Table 2.** (Continued)

| | Competency | | Objective | Consensus Threshold ≥80% | Novice Track | Advanced Track |
|---|---|---|---|---|---|---|
| 5 | **Characteristics of Effective Collaborations and Partnerships**<br><br>**Consensus to keep competency: 94.6%** | a | To identify how successful bidirectional or mutually beneficial engagements function in the context of global operating systems, with inclusion of concepts such as sovereign obligation, transparency, health inequities, global health diplomacy, and cultural humility. | 91.5% | X | X |
| | | b | To understand the practical aspects of identifying and entering bidirectional/ mutually beneficial engagements for improving surgical care (e.g. initiating early communication, MOUs), identifying signs of a failing partnership, and problem-solving when partnership agreements are not upheld. | 89.8% | | X |
| | | c | To identify challenges associated with surgical advocacy and implementation of change with an awareness of cultural context, the sustainability of donor-driven partnerships, imbalances of power in a region outside of one's own background. | 93.2% | X | X |
| 6 | **Ethics and Professionalism in Global Surgery**<br><br>**Consensus to keep competency: 100%** | a | To understand basic ethical principles (beneficence, nonmaleficence, justice, respect for persons, etc.) and define the common ethical variations and challenges unique to delivery of surgical care in diverse cultural/political/economic settings, with an emphasis on practicing within one's skill and experience level. | 98.3% | X | X |
| | | b | To demonstrate understanding of local, national, and international codes of ethics relevant to the surgical and perioperative work environments, including the International Ethical Guidelines for Biomedical Research Involving Human Subjects, and how they influence the practical application of surgical ethical principles in patient care and research. | 96.6% | X | X |
| | | c | To understand the ethical considerations related to neocolonialism and actionable steps that can be taken to achieve the decolonization of surgical and perioperative care, as well as to understand the impact of surgical missions on local communities. | 91.5% | X | X |
| 7 | **Health Equity, Social Justice, and the Right to Essential Surgical Care**<br><br>**Consensus to keep competency: 96.6%** | a | To understand relevant international and regional organizations in linking health and human rights and the Universal Declaration of Human Rights, and how international and national organizations are networking and advocating for local solutions to the challenges in delivering quality health to all based on human rights. | 89.8% | X | X |
| | | b | To define the universal right to access timely, safe, and affordable essential surgery, and demonstrate a basic understanding of the relationship between health, human rights, and global inequities. | 94.9% | X | X |
| | | c | To describe and demonstrate how to implement strategies to engage underserved populations in making decisions that affect their health and well-being. | 86.4% | X | X |
| 8 | **Leadership**<br><br>**Consensus to keep competency: 94.9%** | a | To describe the challenges inherent to being an effective leader in groups with members from various backgrounds and in different contexts (e.g., academic, humanitarian, clinical, and research) within global surgery, with an emphasis on developing local, regional, and national leaders in countries where they are most needed. | 89.8% | | X |
| | | b | To demonstrate the characteristics of an effective mentor to junior members of the surgical and perioperative team and peers within the field of global surgery and to understand how to nurture leadership potential, with an emphasis on fostering mentorship at the local and regional level. | 91.5% | | X |

*(Continued)*

**Table 2.** (Continued)

| | | | Objective | Consensus Threshold ≥80% | Novice Track | Advanced Track |
|---|---|---|---|---|---|---|
| 9 | **Research Equity and Publication**<br><br>**Consensus to keep competency: 96.6%** | a | To understand principles of research ethics regarding conduct of surgical and perioperative research, including the concept of informed consent, and to identify challenges to performing collaborative surgical research with mutual benefit to all parties involved. | 98.3% | X | X |
| | | b | To define the universal right to access timely, safe, and affordable essential surgery, and demonstrate a basic understanding of the relationship between health, human rights, and global inequities. | 94.9% | X | X |
| | | c | To describe and demonstrate how to implement strategies to engage underserved populations in making decisions that affect their health and well-being. | 86.4% | X | X |

*Participant 10 (Round 1, Mozambique): "Competency 8 development should be biased towards Global Surgeons from the LICs and underserved populations."*

*(Competency 8)*

*Participant 18 (Round 2, Rwanda): "Mentoring is not always to junior members of the team. Peer mentoring as a concept should be embedded."*

*(Competency 8)*

*Participant 41 (Round 2, Kenya): "Mentorship in global surgery requires skills/ knowledge of the workings of both the systems in which the mentee finds himself/ herself in." (Competency 8) and "Research in context of global surgery/ health has not always ethical or equitable—this history needs to be acknowledged somewhere, to help generations move forward."*

*(Competency 9)*

Respondents also repeatedly stressed the importance of national health financing and health policy related to surgical systems, such as National Surgical, Obstetric, and Anesthesia Plans (NSOAPs), as well as the importance of local environments and leadership:

*Participant 13 (Round 1, Ecuador): "Inclusion of current local surgical policies and efforts should be taught to provide perspective and understanding of local and global health policies."*

*Participant 18 (Round 1, Pakistan): "To define core principles of healthcare economics and health systems in surgical services delivery and analyze various models of healthcare financing and national health plans and their effects on population health and surgical health expenditure. This is incredibly important. It encompasses the need to know the various frameworks that exist to break down health systems including the NSOAP framework that is based on the WHO building blocks framework but also has space for the control knob framework that is more policy oriented."*

*Participant 20 (Round 2, Kenya): "The entities analyzed/considered should be broadened and should include all non-government entities from NGOs to Parastatals to Trusts to Large grants like NIH that fund "projects" that become entities that impact surgical services/health services and include in that analysis governance and accountability."*

Lastly, acknowledging and dismantling the influences of neocolonialism was considered key to curricula in academic global surgery and perioperative care by the majority of

respondents, though a few either disagreed or were concerned over the politicization of the terminology. These representative quotes demonstrate the most common perspectives noted by participants.

> *Participant 18 (Round 2, Rwanda): "This too is important and timely and getting a lot of attention of late. Again, both novice and advanced learner need to know such information, history, pitfalls, creation of dependency, etc."*

> *Participant 28 (Round 2, Canada): "It might be better to specify that for example learners should understand the cultural influences and historical context of the place where they live or work, it is not feasible to expect them to understand the nuances of every global setting."*

> *Participant 41 (Round 2, Kenya): "An understanding that the current ethical statements are generally based on the Western culture/ values, and the need to be sensitive to/ inclusive of other values within the local contexts in which research may be carried out."*

## Discussion

### Key competencies generating the greatest consensus

After both rounds, the Delphi process resulted in consensus around fundamental competencies in academic global surgery and perioperative care for future learners in both HICs and LMICs. The Delphi process led to a substantial evolution of consensus and prioritization of concepts, language, and focus of this framework. The initial drafts included clinical care and safety- oriented concepts which did not reach consensus and were removed to focus on the broader nature of surgical education as it relates to leadership in surgical systems world-wide. The inclusion of patient care activities was perceived as context specific and there was concern that including them would result in a slippery slope towards the model of HIC trainees visiting to learn from LMIC settings. Furthermore, there was substantial interest in adding Competencies 8 (Leadership) and 9 (Research Equity and Publication) as distinct areas of learning that were necessary for this field. The focus on leadership is particularly consistent with the needs and goals emphasized recently by Hamid et al who noted that inequities in global health leadership need to be addressed through explicit mentorship of the considerably fewer LMIC leaders [23]. Respondents also qualitatively addressed the level of learner (novice vs. advanced) and the core objectives below each competency as shown in Table 2, again using a threshold of 80% consensus on whether each objective was suitable for novice learners only, advanced learners only, or both. Furthermore, the theme of decolonization also arose through the Delphi process.

The competencies with the greatest consensus were:

1. The study of the global burden of surgical disease and the social and environmental determinants of health that contribute to disparities in surgical care in the local, national and international level (Competency 1: Global Burden of Surgical and Perioperative Conditions and Injuries (98.3% consensus) and Competency 3: The Impact of Social and Environmental Determinants of Health and Surgical Care (96.7%))

2. The scholarly examination of surgical ecosystems and relevant ways of strengthening surgical systems with local quality-improvement and capacity building to address the global burden of surgical disease (GBSD) (Competency 4: Strengthening Surgical Systems Capacity (96.6%))

3. The teaching of ethical principles and variations unique to delivering surgical care in diverse cultural, political, and economic settings, with an emphasis on social justice and a human right to surgical care (Competency 6: Ethics and Professionalism in Global Surgery (100%) and Competency 7: Health Equity, Social Justice, and the Right to Essential Surgical Care (96.6%)).

The population-based approaches should focus on public health efforts relevant to global surgery, including epidemiology and public health indicators such as the Sustainable Development Goals (Competency 1) and monitoring and evaluation approaches such as quality improvement initiatives under local leadership (Competency 4). Individual approaches should focus on providing culturally sensitive, ethically conscientious care with a willingness to enter into collaborations with bilateral accountability.

## Prioritizing the LMIC context in competency development

Global health, including global health education, has evolved with an emphasis on HIC institutions, people, and ideas with inadequate representation of the voices of LMIC experts, institutions, and communities despite their expertise in the needs and priorities of their contexts [14–16]. The very definition of global surgery has been derived largely from HIC scholars and maintains that global surgery is "an area for study, research, practice, and advocacy that places priority on improving health outcomes and achieving health equity for all people worldwide who are affected by surgical conditions or have a need for surgical care" and "a synthesis of population-based approaches and individual-level clinical care" [17]. However, this definition fails to include the principles and content that are necessary for this field broadly, and has lacked consistent input from LMIC experts. Unfortunately, a full 95% of formal graduate programs in global health are in HICs, making them unavailable to those in LMICs, where the majority of the world's population lives [18]. This monopoly on global health education exacerbates the already skewed power dynamics across the world [15, 19]. Standards in global health education were developed by Consortium of Universities for Global Health (CUGH) and a consensus-based competency framework was published in 2015. However, many have argued that this CUGH framework did not have sufficient input from LMIC experts and that weakness has led to a narrow focus and unsuitable metrics and resources for assessment [14, 16, 17, 20, 21]. These are important factors for the developing field of academic global surgery to consider in creating educational content that meets the training needs of the next generation. The inherent imbalance in power and equity amongst the global health community, as is increasingly apparent in the academic global surgical literature, will need to be explicitly considered.

As educational programs in global surgery are set up, we must establish a foundation that counterbalances these HIC-centric forces with universally-applicable academic competencies beyond the default HIC-centric viewpoint. Through this project, we intentionally aimed to seek perspectives from around the world to develop a broad and universal set of geography-agnostic competencies centered on the values and principles encountered in LMIC as well as HIC settings. While the logistics of this project were coordinated by members of the American College of Surgeons Operation Giving Back Education Sub-Committee, we intentionally formed a task force of surgeons from around the world who have deep knowledge of various practice environments, extensive expertise in medical and surgical education to govern the framework development, and who have had major leadership positions in surgical societies around the world. This international task force then determined who among their networks would be valuable contributors to this Delphi process. As we have noted in a previous paper, LMIC expert input has to be intentionally included in global surgery education [10]. This

input has been noted as missing in global health education competencies and emphasized by Sayegh et al recently as a critical perspective that needs to be included in all global health education [17]. Several other publications have also recently emphasized this as a mechanism to address the power asymmetries in global health education and promote decolonization [17, 22, 23]. Our intentional inclusion of the LMIC voice and worldview in this Delphi process led to more explicit language regarding decolonization and social justice in the final framework that was not present in our initial framework draft.

## Creating competencies specific to *Academic Global Surgery*

Our findings highlight the core competencies unique to academic global surgery. Global health competencies across clinical disciplines, such as the competencies published by the Consortium of Universities of Global Health (CUGH), have been published already but do not include competencies specific to surgical and perioperative disciplines and historically have not had adequate LMIC input [10]. This Delphi process sought to ensure a global focus especially skewing towards LMICs and addressing competencies needed in the surgical and perioperative disciplines.

Interestingly, some competencies did not meet criteria for inclusion in the final framework. The clinical competency of professional practice was removed after Round 1 as it did not meet the 80% threshold. In some discussions with individual authors, we heard that the diversity of clinical conditions across various settings would limit this competency's generalizability and that including this competency would implicitly lead to a slippery slope that turns the gaze of this framework back to HIC clinicians visiting LMIC settings. Others commented that clinical practice competencies are already outlined by groups such as the Accreditation Council for Graduate Medical Education (ACGME), though not specifically for the LMIC setting, and that devising context-specific clinical competencies would be beyond the scope of this effort. For these reasons, professional practice was removed and this framework was narrowed to academic competencies. A competency on Patient Safety and Quality Improvement also did not meet threshold for inclusion and was also removed during the process. It is possible that implementation of this framework will lead to lessons related to these two competencies that will inform future iterations.

## Future directions for global surgery education

In this paper, we explicitly call for the development of an independent governing body with equitable representation of experts from LMICs and HICs, with defined term limits, to convene and periodically update the framework and integrate lessons from attempts at application and validation. To be effective in creating the next generation of leaders in academic global surgery and perioperative care, this framework has to inform all new and existing global surgical education programs. It must be validated and assessments must be created to determine if learners are indeed acquiring these competencies through educational programs. Furthermore, a free online curriculum must be developed and hosted so that it is widely available worldwide.

This will formalize and standardize training in this evolving field and ensure rigor in training and practice that will ultimately lead to consistent care for our patients. This is not possible without engagement by surgical societies and educational institutions worldwide and is an opportunity for those interested in global surgery and education. The COVID-19 pandemic has resulted in massive changes in education including that of health professionals especially with a focus on technology and societal concerns including health equity [24]. This is an opportune time to harness those lessons and apply them to the development of global surgical education programs.

## Limitations

Although we included various surgical and perioperative specialists in our Delphi process, additional work is needed to expand the concepts underlying these competencies more explicitly and in depth to additional fields, including anesthesia and obstetrics and gynecology, as well as surgical subspecialties pertinent to the unmet global burden of surgical disease, such as ENT, plastic surgery, neurosurgery, and orthopedic surgery and other professions such as nursing and pharmacy. Additionally, despite the best efforts of the logistics team and task force, we did not reach sufficient experts in Australia and the Middle East for input in this project. We also did not attempt to establish assessments of acquiring these competencies through global surgical educational programs as that was beyond the scope of this project. Lastly, this framework has not yet been validated through educational research. All of these limitations need to be addressed to develop a robust academic field covering global surgery and perioperative care.

## Conclusion

We developed a consensus-based set of competencies to inform educational programs in academic global surgery and perioperative care in collaboration with leading and diverse experts in global surgery. Future directions will include developing open access curricula using these competencies, validating them, developing assessments of knowledge acquisition, broadening the process to include other surgical and perioperative subspecialties, and establishing a governing body to oversee revisions of this framework.

## Acknowledgments

We would like to thank Natalie Bell and Miranda Melone for managing the email reminders through the ACS OGB Office.

## Author Contributions

**Conceptualization:** Christine Dart, Hernan Sacoto Aguilar, Emmanuel Ameh, Abebe Bekele, Maria F. Jimenez, Kokila Lakhoo, Doruk Ozgediz, Nobhojit Roy, Sudha Jayaraman.

**Data curation:** Natalie Pawlak, Christine Dart, Hernan Sacoto Aguilar, Emmanuel Ameh, Abebe Bekele, Nobhojit Roy, Girma Terfera, Adesoji O. Ademuyiwa, Barnabas Tobi Alayande, Nivaldo Alonso, Geoffrey A. Anderson, Stanley N. C. Anyanwu, Alazar Berhe Aregawi, Soham Bandyopadhyay, Tahmina Banu, Alemayehu Ginbo Bedada, Anteneh Gadisa Belachew, Fabio Botelho, Emmanuel Bua, Leticia Nunes Campos, Chris Dodgion, Michalina Drejza, Marcel E. Durieux, Rohini Dutta, Sarnai Erdene, Rodrigo Vaz Ferreira, Zipporah Gathuya, Dhruva Ghosh, Randeep Singh Jawa, Walter D. Johnson, Fauzia Anis Khan, Fanny Jamileth Navas Leon, Kristin L. Long, Jana B. A. Macleod, Anshul Mahajan, Rebecca G. Maine, Grace Zurielle C. Malolos, Craig D. McClain, Mary T. Nabukenya, Peter M. Nthumba, Benedict C. Nwomeh, Daniel Kinyuru Ojuka, Norgrove Penny, Martha A. Quiodettis, Jennifer Rickard, Lina Roa, Lucas Sousa Salgado, Lubna Samad, Justina Onyioza Seyi-Olajide, Martin Smith, Nichole Starr, Richard J. Stewart, John L. Tarpley, Julio L. Trostchansky, Ivan Trostchansky, Thomas G. Weiser, Adili Wobenjo, Elliot Wollner, Sudha Jayaraman.

**Formal analysis:** Natalie Pawlak, Christine Dart, Sudha Jayaraman.

**Investigation:** Natalie Pawlak, Christine Dart, Sudha Jayaraman.

**Methodology:** Christine Dart, Emmanuel Ameh, Maria F. Jimenez, Kokila Lakhoo, Doruk Ozgediz, Girma Terfera, Sudha Jayaraman.

**Supervision:** Sudha Jayaraman.

**Writing – original draft:** Natalie Pawlak, Christine Dart, Sudha Jayaraman.

**Writing – review & editing:** Natalie Pawlak, Christine Dart, Hernan Sacoto Aguilar, Emmanuel Ameh, Abebe Bekele, Maria F. Jimenez, Kokila Lakhoo, Doruk Ozgediz, Nobhojit Roy, Girma Terfera, Adesoji O. Ademuyiwa, Barnabas Tobi Alayande, Nivaldo Alonso, Geoffrey A. Anderson, Stanley N. C. Anyanwu, Alazar Berhe Aregawi, Soham Bandyopadhyay, Tahmina Banu, Alemayehu Ginbo Bedada, Anteneh Gadisa Belachew, Fabio Botelho, Emmanuel Bua, Leticia Nunes Campos, Chris Dodgion, Michalina Drejza, Marcel E. Durieux, Rohini Dutta, Sarnai Erdene, Rodrigo Vaz Ferreira, Zipporah Gathuya, Dhruva Ghosh, Randeep Singh Jawa, Walter D. Johnson, Fauzia Anis Khan, Fanny Jamileth Navas Leon, Kristin L. Long, Jana B. A. Macleod, Anshul Mahajan, Rebecca G. Maine, Grace Zurielle C. Malolos, Craig D. McClain, Mary T. Nabukenya, Peter M. Nthumba, Benedict C. Nwomeh, Daniel Kinyuru Ojuka, Norgrove Penny, Martha A. Quiodettis, Jennifer Rickard, Lina Roa, Lucas Sousa Salgado, Lubna Samad, Justina Onyioza Seyi-Olajide, Martin Smith, Nichole Starr, Richard J. Stewart, John L. Tarpley, Julio L. Trostchansky, Ivan Trostchansky, Thomas G. Weiser, Adili Wobenjo, Elliot Wollner, Sudha Jayaraman.

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
