## [Decision Letter · Decision Letter 0]

4 Apr 2023

PGPH-D-23-00129

Academic Global Surgical Competencies: A Modified Delphi Consensus Study

Dear Dr. Jayaraman,

Thank you for submitting your manuscript to PLOS Global Public Health. After careful consideration, we feel that it has merit but does not fully meet PLOS Global Public Health’s publication criteria as it currently stands. Therefore, we invite you to submit a revised version of the manuscript that addresses the points raised during the review process.

In addition to addressing the comments from the reviewers, please also combine Figure 3 into Table 1.

We look forward to receiving your revised manuscript.

Kind regards,

Bethany Hedt-Gauthier, PhD

Academic Editor

Journal Requirements:

Additional Editor Comments (if provided):

Reviewers' comments:

Reviewer's Responses to Questions

**Comments to the Author**

1. Does this manuscript meet PLOS Global Public Health’s publication criteria? Is the manuscript technically sound, and do the data support the conclusions? The manuscript must describe methodologically and ethically rigorous research with conclusions that are appropriately drawn based on the data presented.

Reviewer #1: Partly

Reviewer #2: Yes

2. Has the statistical analysis been performed appropriately and rigorously?

Reviewer #1: N/A

Reviewer #2: Yes

3. Have the authors made all data underlying the findings in their manuscript fully available (please refer to the Data Availability Statement at the start of the manuscript PDF file)?

Reviewer #1: Yes

Reviewer #2: Yes

4. Is the manuscript presented in an intelligible fashion and written in standard English?

Reviewer #1: Yes

Reviewer #2: Yes

5. Review Comments to the Author

Reviewer #1: Pawlak et al. present a valuable research endeavor. I believe the article will be of interest to the PLOS Global Public Health readership given its robust methodology and insightful results. Another important strength of this publication is its inclusion of diverse global surgery experts. However, I recommend against publication of the manuscript in its form given the concerns enumerated below:

Abstract:

- Background, First sentence: I suggest the authors focus on academic global surgery rather than global surgery as a whole. The reason being that the skills required for academic global surgery will most likely be different from those needed for education, policy implementation, advocacy, or entrepreneurship in the form of reverse and frugal innovation. Based on the same rationale and if the word limits allow it, I suggest adding "Academic" to the short title.

- Background, second sentence: This sentence is not self-explanatory. It is unclear which competencies the authors referring to.

- Background, Third sentence: The purpose of the Delphi is not "to develop competencies" but rather to determine which competencies are essential. Please, edit the third sentence to reflect this.

- Results, First sentence: This sentence is a restatement of the first sentence in the methods. The only difference being the number of countries represented. Either focus on the methodology or criteria used to select the task force members in the methods section or use the results section sentence in place of the first sentence in the methods. Also, "Diverse" is a tautology here since diversity is exemplified by the number of countries represented.

- Conclusion, First sentence: It is more accurate to say the Delphi "identified" nine competencies rather than "led to." The authors mention these skills will lead to the development of standardized academic global surgery curricula. This is the first time this curriculum is mentioned. The background seemed to focus on academic global surgery practice rather than academic global surgery education.

Introduction

- The first paragraph focuses on global surgery in general. The next paragraph discusses global surgery education and academic global surgery and concludes that there is a need for standardization. As it stands, these paragraphs fail to answer a few key questions. It is perplexing that the authors do not define global surgery or academic global surgery in the first paragraph but use these terminologies only to state that there is no unified definition of global surgery. While this statement is true, it is essential that the authors state under which definitions of global surgery and academic global surgery they are working. These definitions are critical since the authors seek "to standardize academic global surgery competencies." If the participants did not agree on a definition, it casts doubt on the Delphi process results. The definitions are even more critical because it appears the authors equate global surgery with academic global surgery and global surgery education. If we assume that the term "Academic" in academic global surgery serves the same role as it does in "Academic Medicine," can we assume that all global surgery practice is academic? Where would this assumption leave civil society organizations like the Global Surgery Foundation, G4 Alliance, surgical non-governmental organizations, etc? As a reminder, these questions are not about being right or wrong but rather they help the reader put the study methodology and findings in context.

- The aforementioned lack of clarification has obscured the study's justification. In lines 79-80 the authors conclude that there is a need for standardized curricula on the fundamentals of academic global surgery based on a review of HIC training material. Are we therefore to understand that the aim of the study was the identification of competencies for the education of aspiring global surgeons in HICs?

- Lines 82-94: This paragraph would probably be better placed in the discussion were it could expand on the importance of ethics and standardization. In its current location, it fails to build on previous themes discussed

- Line 118: Instead of "diverse" please elaborate on some of the criteria that were considered. Perhaps the committee considered, income country levels, WHO or World Bank region, surgical and peri-operative specialty, seniority, gender, non-physician professionals, etc. The idea here is to show and not tell thereby allowing the readers to come to the same conclusion as the authors.

- Lines 159-165: Please, define "core competency" and "sub-competency."

- Discussion: Please, put these competencies in context with regards to existing global surgery MSc/PhD programs, research fellowships, and other training program curricula.

Reviewer #2: Thank you for an opportunity to review this work. Overall this is very well written and easy to follow. I especially appreciate the inclusion of a majority of LMIC consensus participants.

I have a few major comments and some minor comments.

Major comments:

1. A Delphi consensus is usually conducted in at least three rounds with the first round including qualitative open ended questions. Please comment why you chose to perform a modified Delphi consensus, provide references for your methodology, and describe much more clearly the process by which you created the first round instrument. In the modified Delphi technique the first round instrument is usually based on literature or preliminary research if it is structured, though it seems like this one was based on consensus of a smaller group of experts? How might that have influenced your findings? Could you comment specifically on how even the notion of global surgery may reflect a colonizing mindset? Is there any possibility that surgeons from LMICs may not even agree that there is or should be such a thing as academic global surgery and that the entire premise for the consensus is flawed? If you don't agree with this possibility that is fine, but I think it should be addressed. Again related to the methodology, how did you decide on two rounds and were there stopping criteria? Why the 5 point Likert scale?

2. How did you determine the sample size? This is large for a Delphi study.

3. While conceptually, I think this research adds value to our general understanding of what competencies should be in academic global surgery (and maybe even more so about what a group of experts think academic global surgery actually is), I'm struck by how most of these competencies apply more generally to academic activities in global health. I think it would be helpful if the authors could provide a discussion of the competencies identified by the Delphia panel related to competencies more generally in global health. Do those exist? Is there consensus there? How are these similar or different? What is the value for having distinct competencies in global surgery?

4. It strikes me as especially noteworthy that competencies regarding clinical practice were removed during the consensus process. This deserves more emphasis, especially in the larger debate around what academic global surgery is and its vision, etc. Are you saying there is minimal or no role for clinical care and clinical training in academic global surgery? That is certainly a major departure from US-based global surgery practice.

5. A significant volume of surgery is provided by non-surgeons in LMICs. Did you think of including their perspectives? Why/why not? Is this a limitation?

Smaller points:

1. Abstract-Background, "inform in" appears to be a typo

2. Line 54, sentence starting in 2015 needs to be reworked

3. Same for the First sentence in the Methods, it is very confusing.

Thank you again for your work and allowing me to review it.

6. PLOS authors have the option to publish the peer review history of their article (what does this mean?). If published, this will include your full peer review and any attached files.

**Do you want your identity to be public for this peer review?** For information about this choice, including consent withdrawal, please see our Privacy Policy.

Reviewer #1: **Yes: **Ulrick Sidney Kanmounye

Reviewer #2: No

---

## [Decision Letter · Decision Letter 1]

5 Jun 2023

Academic Global Surgical Competencies: A Modified Delphi Consensus Study

PGPH-D-23-00129R1

Dear Dr. Jayaraman,

We are pleased to inform you that your manuscript 'Academic Global Surgical Competencies: A Modified Delphi Consensus Study' has been provisionally accepted for publication in PLOS Global Public Health.

Best regards,

Bethany Hedt-Gauthier, PhD

Academic Editor

The reviewers/editor are overall satisfied with the team's thoughtful edits based on the last review. I do ask that the authors go back to R2's comment on the lack of inclusion of non-clinicians. 1) I don't entirely agree with the comment that these individuals are not engaged in academic global surgery, and 2) if that is true, then that is a limitation of the field. Either or both the limitation of the field or of the study should be addressed in the limitations section.

Reviewer Comments (if any, and for reference):

Reviewer's Responses to Questions

**Comments to the Author**

1. If the authors have adequately addressed your comments raised in a previous round of review and you feel that this manuscript is now acceptable for publication, you may indicate that here to bypass the “Comments to the Author” section, enter your conflict of interest statement in the “Confidential to Editor” section, and submit your "Accept" recommendation.

Reviewer #2: All comments have been addressed

2. Does this manuscript meet PLOS Global Public Health’s publication criteria? Is the manuscript technically sound, and do the data support the conclusions? The manuscript must describe methodologically and ethically rigorous research with conclusions that are appropriately drawn based on the data presented.

Reviewer #2: Yes

3. Has the statistical analysis been performed appropriately and rigorously?

Reviewer #2: Yes

4. Have the authors made all data underlying the findings in their manuscript fully available (please refer to the Data Availability Statement at the start of the manuscript PDF file)?

Reviewer #2: Yes

5. Is the manuscript presented in an intelligible fashion and written in standard English?

Reviewer #2: Yes

6. Review Comments to the Author

Reviewer #2: Thank you for making the appropriate edits based on reviewer feedback. I have no further comments.

7. PLOS authors have the option to publish the peer review history of their article (what does this mean?). If published, this will include your full peer review and any attached files.

**Do you want your identity to be public for this peer review?** For information about this choice, including consent withdrawal, please see our Privacy Policy.

Reviewer #2: No
